# Coarse Correspondences Boost 3D Space-time Understanding in Multimodal Language Model

## Abstract

Multimodal language models (MLLMs) are increasingly being applied in real-world environments, necessitating their ability to interpret 3D spaces and comprehend temporal dynamics. Current methods often rely on specialized architectural designs or task-specific fine-tuning to achieve this. We introduce Coarse Correspondences, a simple lightweight method which enhances MLLMs' understanding of 3D and temporal concepts using only 2D images, without modifying the architecture or task-specific fine-tuning. Our method uses a lightweight tracking model to identify primary object correspondences between frames in a video or across different image viewpoints, and then conveys this information to MLLMs through visual prompting. We demonstrate that this simple training-free approach brings substantial gains to GPT4-V/O consistently on four benchmarks that require 3D and temporal understanding, including **+20.5%** improvement on ScanQA, **+9.7%** on OpenEQA's episodic memory subset, **+6.0%** on the long-form video benchmark EgoSchema, and **+11%** on the R2R navigation benchmark. Additionally, we show that Coarse Correspondences can also enhance open-source MLLMs' understanding of 3D space (by **+6.9%** on ScanQA) when applied in both training and inference and that the improvement can generalize to unseen datasets such as SQA3D (**+3.1%**). Taken together, we show that Coarse Correspondences effectively and efficiently boosts models' performance on downstream tasks requiring 3D and/or temporal understanding.

## 1 Introduction

Intelligence is multi-faceted. While multi-modal large language models (OpenAI, 2024) have shown remarkable linguistic, logical and even mathematical intelligence, many remain doubtful about their visual and spatial intelligence. Despite their excellent performance on visual-lingusitic tasks, many recent works (Majumdar et al., 2024; Singh et al., 2024) demonstrate that state-of-the-art MLLMs still struggle at 3D and long video benchmarks, performing only marginally better than blind text-only baselines. These results suggest that 3D and temporal understanding are two major bottlenecks on MLLMs' path to general visual intelligence.

To enhance MLLMs' 3D understanding, researchers have mainly explored three approaches: providing MLLMs with 3D data as input (Hong et al., 2024), designing specialized architectures for 3D tasks (Hong et al., 2024), or employing supervised fine-tuning with 3D data (Chen et al., 2024). Similarly, to boost MLLMs' temporal understanding, prior works have proposed new model architectures designed for long video understanding (Papalampidi et al., 2023; Balažević et al., 2024), or adopted Socratic-based methods (Zhang et al., 2024; Kahatapitiya et al., 2024) (i.e., converting each frame of a video into text using a caption model, and then using text-only LLMs to summarize).

In contrast to prior works, we propose a simple but effective training-free visual prompting method Coarse Correspondences to boost 3D and temporal understanding in MLLMs. Coarse Correspondences uses a tracking model to extract object-level correspondences across multiple images, and then represent the most salient correspondence relationships on the images through visual prompting. Our method significantly boosts MLLMs' understanding of 3D spacetime with only 2D image inputs and without any specialized architectural design or task-specific fine-tuning.

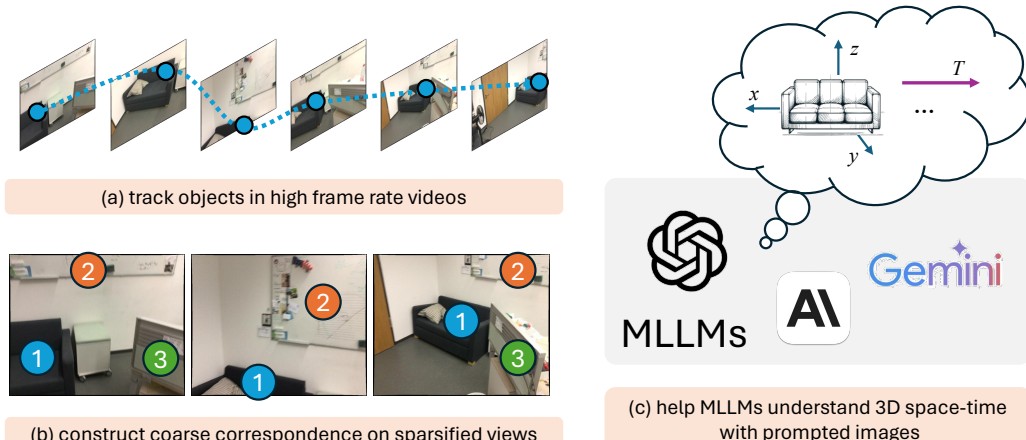

(a) track objects in high frame rate videos

(b) construct coarse correspondence on sparsified views

(c) help MLLMs understand 3D space-time with prompted images

Figure 1: We combined light-weight video tracking models and multimodal LLMs to achieve a better understanding of 3D spacetime. (a) We use a tracking model at a high frame rate to obtain instance segmentation masks for each frame. (b) Then, we sequentially sparsify input frames, select prominent coarse correspondences, and visualize the constructed coarse correspondences on the images. (c) Finally, we enable MLLMs to better understand 3D spacetime from the prompted images.

We have demonstrated substantial performance gains of COARSE CORRESPONDENCES through extensive experiments with both open-source and closed-source models across 6 benchmarks on 3D spatial understanding and temporal understanding. For closed-source models, we apply COARSE CORRESPONDENCES on GPT4-V/O during inference and achieve compelling gains. First, on 3D understanding, we show that our method significantly surpasses state-of-the-art models by 20.5% and 9.7% on ScanQA (Azuma et al., 2022) and OpenEQA (Majumdar et al., 2024) respectively. Second, for long video understanding, our method leads to a 6% gain in performance on the EgoSchema benchmark (Mangalam et al., 2023). Notably, our method uses much fewer input images and, in a zero-shot manner, outperforms many fine-tuned models that use far more images. For example, on EgoSchema, COARSE CORRESPONDENCES surpasses state-of-the-art results with just 8 uniformly sampled frames from a 3-minute video, greatly reducing the computational costs of MLLMs compared to existing methods. In addition to 3D and video QA tasks, we further demonstrate that our method enhances models' performance on embodied tasks such as navigation (Krantz et al., 2020), which require strong spatial and temporal understanding, by 11% in success rate on R2R. These results suggest that COARSE CORRESPONDENCES boosts MLLMs' spatial and temporal understanding both effectively and efficiently. Last but not least, we experiment with open-source MLLMs (Liu et al., 2024a) by applying COARSE CORRESPONDENCES in both instruction tuning and inference; again, our method shows performance gains against the baseline (by 6.9% on ScanQA), and the improvement even generalizes to unseen datasets such as SQA3D (+3.1%). These results suggest that COARSE CORRESPONDENCES works well universally with any model – both closed-source and open-source – that can take in multiple images and understand visual markers.

To further understand why our simple method brings significant improvements on spatial and temporal understanding, we conducted additional investigations with a small diagnostic benchmark on spatial orientation, as orientation is a key component of spatial understanding and important to navigation tasks. We manually curated a benchmark called SOT to test how well MLLMs understand left-right relationships from different viewpoints on images taken by cameras with different motions. Our experiments reveal that (1) even GPT-4o struggles with understanding images taken by a camera moving right to left (instead of the more common motion left to right) and (2) that COARSE CORRESPONDENCES leads to improvements on this benchmark. These results indicate that COARSE CORRESPONDENCES mitigates the camera motion bias in MLLMs – their tendency to understand images better when the camera moves in a particular direction – when processing visual signals and helps them learn a more equivariant internal visual representation.

Overall, we want to highlight with this work that, despite its simplicity and being underestimated for semantic tasks in deep learning, visual correspondence can still bring significant utility to spatial and temporal understanding in MLLMs, just as it has long contributed to 3D reconstruction Schonberger & Frahm (2016). Although MLLMs still exhibit some non-negligible shortcomings in spatial

and temporal understanding, even with COARSE CORRESPONDENCES —such as the ability to perform spatial perspective-taking—we hope our work demonstrates the potential of leveraging visual correspondences to help MLLMs better understand our physical world.

## 2 METHOD

We introduce COARSE CORRESPONDENCES, a visual prompting method that allows MLLMs to reason about 3D space and time.

**Problem formulation.** Given a question $\mathcal{Q}$ and a sequence or set of observations in an environment $[I_1, \ldots, I_n]$, our aim is to design a visual prompt $\mathcal{P}(\ldots)$ that modifies the input image set. These image inputs don't have to be a video. They can also represent a set of images of a specific scene from multiple viewpoints. We evaluate the prompt by measuring its utility in prompting an MLLM $\mathcal{M}$:

$$[I'_1, \ldots, I'_n] = P([I_1, \ldots, I_n])$$
$$\hat{\mathcal{A}} = \mathcal{M}(([I'_1, \ldots, I'_n]), \mathcal{Q})$$

We compare the generated answer $\hat{\mathcal{A}}$ with the ground truth $\mathcal{A}$.

In our framework, the MLLMs can be any general-purpose model without requiring any special architecture or training for 3D or temporal understanding. Our aim is to develop a prompting strategy that allows models to improve such capabilities without any training (Figure 1).

COARSE CORRESPONDENCE

Our prompting method, COARSE CORRESPONDENCES, contains four steps: (1) tracking correspondences, (2) sparsify frames, (3) selecting, and (4) visualizing coarse correspondences.

**(1) Tracking correspondences.** Given $n$ input images, $[I_1, \ldots, I_n]$, we first use an off-the-shelf video object tracking model, such as Tracking Anything (Yang et al., 2023c). This model extracts class-agnostic instance segmentation masks $(M_1, \ldots, M_n)$ for each image. Each $M_i$ is a $H \times W$ dimensional matrix where $H$ and $W$ are the height and width of the input image $I_i$. Each pixel location in $M_i$ contains an instance ID, indicating which instance the pixel at that position belongs to within the image sequence.

**(2) Sparsify frames.** Since most MLLMs contain a large number of parameters, directly using them to process long image sequences is very computationally intensive. Additionally, proprietary MLLMs like GPT-4O can also incur significant costs if the number of image tokens that need to be processed increases. Reducing the number of input images might lose vital information necessary for MLLMs.

COARSE CORRESPONDENCES strikes a balance in this tradeoff by extracting meaningful video object tracks (a relatively cheaper operation) from high-frame-rate image sequences, and then samples a few image inputs along with the tracks, to retain—and even improve—performance while reducing the MLLM's computation cost. From this extracted video object tracks, we perform temporal downsampling, retaining only $m << n$ uniformly sampled images and their corresponding masks, denoted as $[I_{s_1}, \ldots, I_{s_m}]$ and $[M_{s_1}, \ldots, M_{s_m}]$, where $s_i \in \{1, \ldots, n\}$. This downsampling reduces the number of images we feed into $\mathcal{M}$.

**(3) Selecting coarse correspondences.** Prompting an MLLM with all the detected correspondences results in information overload. In fact, our ablations (discussed in Sec 5) find that adding all the correspondences reduces the MLLM's performance. Therefore, we select a subset of prominent instances to retain. We select the prominent instances of the top-K objects that co-occur in the most number of frames. We first calculate the occurrence frequency and area sum of each unique instance ID in the retained $m$ masks using the following equation:

$$\mathcal{F}req(\text{ID}) = \sum_{i=s_1}^{s_m} \mathbf{1}_{\{\text{ID} \in M_i\}},$$

$$\mathcal{A}rea(\text{ID}) = \sum_{i=s_1}^{s_m} \sum_{p \in M_i} \mathbf{1}_{\{\text{ID}=p\}}.$$

| Model | Frame | BLEU-1 | BLEU-2 | METEOR | ROUGE-L | CIDEr |
|---|---|---|---|---|---|---|
| *3D-Specific Models* | | | | | | |
| ScanQA (Azuma et al., 2022) | - | 26.9 | 16.6 | 11.5 | 30 | 55.4 |
| ScanRefer+MCAN (Yu et al., 2019) | - | 30.2 | 20.4 | 13.1 | 33.3 | 64.9 |
| 3D-LLM (Hong et al., 2024) | - | 39.3 | 25.2 | 14.5 | 35.7 | 69.4 |
| *Open-source Multi-modal Models* | | | | | | |
| LLaVA(Fine-tuned) | 64 | 34.7 | 22.0 | 13.8 | 31.1 | 67.3 |
| LLaVA+*Coarse Correspondences* | 64 | **38.6** | **24.7** | **15.4** | **38.3** | **74.2** |
| *Proprietary Multi-modal Models* | | | | | | |
| GPT-4V | 8 | 28.6 | 13.4 | 13.5 | 33.4 | 59.6 |
| GPT-4V+*Coarse Correspondences* | 8 | **39.7** | **25.5** | **17.4** | **40.8** | **79.2** |
| GPT-4O | 4 | 30.5 | 19.8 | 14.8 | 36.1 | 72.2 |
| GPT-4O+*Coarse Correspondences* | 4 | **35.4** | **25.5** | **18.0** | **42.6** | **87.0** |

Table 1: **Comparison on ScanQA validation set.** We conduct experiments on the ScanQA validation set to demonstrate the effectiveness of COARSE CORRESPONDENCES with different MLLMs. Our method enables both proprietary models and open-source models to surpass all 3D-specific models.

Then, we first sort all instance IDs in descending order based on $\mathcal{F}req(\text{ID})$. If there are ties, we further sort based on $\mathcal{A}rea(\text{ID})$. Finally, we retain the top $k$ instance IDs as tracklets, denoted as $[T_1, \ldots, T_k]$, to visualize for MLLMs.

**(4) Visualizing coarse correspondences.** For each set of obtained correspondence relationships, we visualize the correspondences directly in the image as a marker. Specifically, for each identified primary instance ID $T_i$, if it exists in the mask $M_{s_j}$ of a retained image $I_{s_j}$, we overlay a mark with a fixed size and shape labeled with $T_i$ at the position $(\bar{x}_{ij}, \bar{y}_{ij})$ on $I_{s_j}$ to produce $I'_{s_j}$. The specific placement position can be easily obtained by:

$$(\bar{x}_{ij}, \bar{y}_{ij}) = \frac{\sum_{(x,y)}(x, y) \cdot \mathbf{1}_{\{M_{s_j}(x,y) = T_i\}}}{\sum_{(x,y)} \mathbf{1}_{\{M_{s_j}(x,y) = T_i\}}}$$

Naturally, we can overlay not just the markers but also the segmentation outlines or even the segmentation masks associated with each retained prominent instance. We explore these ablations later. In the end, we obtain the prompted image sequence $[I'_1, \ldots, I'_m]$, which is then used as the input to MLLMs.

We refer to our method as *Coarse* because of the following: first, we only visually prompt for instance-level correspondences and not point-level correspondences. Second, the instance-level correspondences are extracted using off-the-shelf tracking models. Despite not being perfectly precise, they still help MLLMs build a better 3D model of the environment. Third, we only visualize a handful of prominent corresponding instances.

## 3 PROMPTING PROPRIETARY MODELS

We first evaluated the utility of our COARSE CORRESPONDENCES on multiple tasks using proprietary models, including understanding 3D space (ScanQA (Azuma et al., 2022) and OpenEQA (Majumdar et al., 2024) in §3.1) as well as temporal events (EgoSchema (Mangalam et al., 2023) in §3.2). Building on the improvements our method brings to 3D understanding and long video understanding, we further demonstrate that our method also delivers significant gains in navigation tasks (VLN-CE ()). Across all these benchmarks, we augment proprietary MLLMs (e.g., GPT-4V and GPT-4O) with COARSE CORRESPONDENCES and evaluate its zero-shot performance. We show that COARSE CORRESPONDENCES significantly improves the base GPT models and can substantially surpass many current state-of-the-art methods that have undergone specialized fine-tuning, even while using much fewer images as input. All experiments were conducted using A100 80G GPUs.

## 3.1 SPATIAL UNDERSTANDING

**Benchmarks**. The validation set of ScanQA dataset contains 4675 questions about 71 scenes. Questions in ScanQA require basic recognition, 3D localization, and 3D embodied capabilities (Duan et al., 2022). The validation set contains two ground-truth answers per question for evaluation with models that produce free-form answers. OpenEQA Dataset is an open-vocabulary dataset benchmarking spatial environment understanding and embodied reasoning. We evaluate on OpenEQA's EM-EQA data split, which contains over 1600 high-quality human-generated questions. The subset tests the episodic memory of an agent moving through a 3D environment over time.

**Baselines.** For ScanQA, we evaluate COARSE CORRESPONDENCES by augmenting both GPT-4{V,O}, Gemini and Claude models. Besides, we also consider 3D specialized models (Yu et al., 2019; Azuma et al., 2022; Hong et al., 2024) fine-tuned on ScanQA. For OpenEQA, we compare against language-only models to account for language bias (LLaMA2 (Touvron et al., 2023)), commonly used general-purpose multimodal LLMs (GPT-4 (OpenAI et al., 2024), Claude3 (Anthropic, 2024), Gemini-Pro (Team et al., 2024), GPT-4V with 15 and 50 frames.

**Metrics.** For ScanQA, following prior works, we adopt BLEU (Papineni et al., 2002) scores, METEOR (Banerjee & Lavie, 2005), ROUGE-L (Lin, 2004), and CIDEr (Vedantam et al., 2015) as our evaluation metrics. For OpenEQA, we follow their evaluation approach by using GPT-4 to compare the generated answers with the ground-truth answers and assign a score. We report the average score across all questions.

**Results.** For ScanQA, as shown in Table 1, compared to raw input, COARSE CORRESPONDENCES consistently improves the overall performance of different proprietary models. For instance, on the strongest model, GPT-4o, COARSE CORRESPONDENCES brings improvements of 5.7 BLEU-2, 3.2 METEOR, 6.5 ROUGE-L, and 15 CIDEr points. Compared to methods that are specifically designed for 3D understanding tasks, fine-tuned with specialized 3D SFT data, or even those that use 3D point clouds instead of 2D images as input, we observe that a general-purpose MLLM can still outperform them, especially when enhanced with COARSE CORRESPONDENCES. Moreover, we found that this can be achieved using far fewer images as input.

We also demonstrated the same conclusion on OpenEQA, as indicated in Table 2. By applying COARSE CORRESPONDENCES, we significantly improved the performance of both GPT-4v and GPT-4o, achieving better results with fewer input images. These findings suggest that general-purpose MLLMs are indeed capable of understanding 3D space, and COARSE CORRESPONDENCES can significantly enhance their spatial understanding while simultaneously reducing the number of views needed, which could lower the inference cost and make MLLMs more useful for embodied tasks.

| Models | Frame | Accuracy |
|---|---|---|
| LLaMA2 (Touvron et al., 2023) | 0 | 28.3 |
| GPT-4 (OpenAI et al., 2024) | 0 | 33.5 |
| Claude3 (Anthropic, 2024) | 20 | 36.3 |
| Gemini-Pro (Team et al., 2024) | 15 | 44.9 |
| GPT-4V (OpenAI, 2023) | 15 | 54.6 |
| GPT-4V (OpenAI, 2023) | 50 | 55.3 |
| Human | Full | 86.8 |
| GPT-4V | 8 | 44.8 |
| GPT-4V+CC | 8 | **58.5** |
| GPT-4O | 4 | 49.4 |
| GPT-4O+CC | 4 | **59.1** |

| Models | Frame | Subset |
|---|---|---|
| LongViViT (Papalampidi et al., 2023) | 256 | 56.8 |
| MC-ViT-L (Balažević et al., 2024) | 128+ | 62.6 |
| LLoVi (Zhang et al., 2024) | 180 | 58.3 |
| VideoAgent (Wang et al., 2024) | 8.4 | 60.2 |
| MVU (Ranasinghe et al., 2024) | 16 | 60.3 |
| VideoAgent (Fan et al., 2024) | - | 62.8 |
| LangRepo (Kahatapitiya et al., 2024) | | 66.2 |
| GPT-4V | 8 | 64.2 |
| GPT-4V+CC | 8 | **67.4** |
| GPT-4O | 8 | 67.2 |
| GPT-4O+CC | 8 | **73.2** |

Table 2: **Comparisons on EM-EQA setting of OpenEQA .** Our method further enhances the embodied ability of MLLMs and exceeds previous methods by a large margin.

Table 3: **Comparisons on EgoSchema validation set.** COARSE CORRESPONDENCES improves existing MLLMs and surpasses previous finetuned models in a zero-shot manner.

## 3.2 TEMPORAL UNDERSTANDING

**Benchmarks.** We evaluated the improvements of our method for long video understanding using the EgoSchema dataset. Each video in EgoSchema is 3 minutes long, with a corresponding question that includes five multiple-choice options. These questions are designed to ensure that answering them

requires viewing a sufficient number of frames from the video. Due to budget constraints, we limited our evaluation to 500 questions from the validation set.

**Baselines.** The baseline methods we compared against include newly designed and trained model architectures specifically for long video understanding, such as LongViViT (Papalampidi et al., 2023) and MC-ViT-L (Balažević et al., 2024). On the other hand, we also compared methods that rely on text-only foundation models (e.g., GPT-4), i.e., Socratic-based approaches (Zhang et al., 2024; Kahatapitiya et al., 2024), which first use an off-the-shelf image captioning model (Zhao et al., 2023) to convert video frames into captions, and then prompt GPT-4 to answer questions based on those captions. Additionally, we compared agent-based methods (Wang et al., 2024; Fan et al., 2024), which involve using GPT-4 alongside an image captioning model in an agent framework to iteratively perform a series of multi-step reasoning operations to understand long videos. In contrast to these approaches, our method is entirely based on an end-to-end general MLLM architecture, exploring how to further enhance its ability to understand long videos.

**Results.** COARSE CORRESPONDENCES demonstrates state-of-the-art performance, significantly outperforming existing approaches in a zero-shot manner (Table 3). Compared to the original GPT-4o model, our method improves its performance by 6%. Notably, our method uses far fewer frames than other approaches, yet achieves higher results compared to methods that use many more frames. It is also worth highlighting that even the original GPT-4o, when limited to just 8 frames, already serves as a very strong baseline. This indicates the potential of a sufficiently powerful general-purpose MLLM in long video understanding.

| Methods | Success Rate ↑ | Oracle Success Rate ↑ | Success weighted by Path Length ↑ | Trajectory Length ↑ | Navigation Error ↓ |
|---------|----------------|------------------------|-----------------------------------|---------------------|--------------------|
| GPT-4O | 12.00 | 18.00 | 10.37 | 7.31 | 8.49 |
| GPT-4O+CC | **23.00** | **29.00** | **21.03** | **8.12** | **7.37** |

Table 4: **Comparison on Navigation task.** COARSE CORRESPONDENCES improves GPT-4o's performance on R2R dataset for different evaluation metrics. Except for NE, where a lower value indicates better performance, higher values for the other metrics reflect better performance.

### 3.3 NAVIGATION

Building on the improvements in 3DQA and VideoQA, we hope that our method can also prove effective in embodied tasks such as navigation. Navigation requires an agent to understand 3D space, such as being able to determine the spatial relationship between objects in the instruction and itself, while also performing temporal reasoning to assess the progress toward completing the instruction. We consider conducting experiments on the VLN-CE benchmark (Krantz et al., 2020), which is a continuous simulation environment for low-level action execution in indoor scenes.

**Setup.** We adopt the val-unseen split from R2R (Krantz et al., 2020) for evaluation. Unlike the previous QA tasks, where all images could be processed at once, in navigation tasks, each image is processed in an online fashion. Specifically, we feed in one image at each iteration of the conversation. Given the significant variation in viewpoints during navigation, we use SAMv2 (Ravi et al., 2024), the state-of-the-art model for long-range tracking, to label each new input image based on episodic history. Then, using the prompted images, we induce the MLLM to output one of four actions at each step: FORWARD (distance), TURN-LEFT (rotation angle), TURN-RIGHT (rotation angle), and STOP. We follow NavGPT (Zhou et al., 2024) to craft input prompts. Considering the high computational cost of navigation tasks, we selected 100 samples from the val-unseen split. Our primary goal is to demonstrate that our method can enhance GPT models' capabilities in zero-shot navigation tasks, which remains a significant challenge for various types of models.

**Metrics.** We follow the standard VLN evaluation metrics to evaluate the navigation performance, including success rate (SR), oracle success rate (OS), success weighted by path length (SPL), trajectory length (TL), and navigation error from goal (NE). Note that an episode is considered successful if the agent calls the STOP action within 3m of the goal in the VLN-CE.

**Results.** As shown in Table 4, our method achieved improvements across all metrics. It can be observed that while GPT-4o performs impressively on many QA tasks, its zero-shot performance

on navigation tasks is relatively low. This may partly be due to the lack of specialized training on action data, making it less accurate in outputs such as determining how many meters to move forward. However, our method reveals another dimension of the problem: MLLMs' understanding of the 3D spacetime in which they operate can be further enhanced. This is evidenced by the significant improvements in navigation when using COARSE CORRESPONDENCES. We believe that our approach holds great potential for embodied tasks, which can be explored in future research.

## 4 PROMPTING OPEN MODELS

We further validate the effectiveness of our COARSE CORRESPONDENCES on open-source models. Our primary goal is to demonstrate that our method is not only effective for powerful proprietary models but also provides general improvements to a wide range of MLLMs. We start with the LLaVA model (Liu et al., 2024a) and fine-tune it using a dataset comprising approximately 1.2 million samples of image and video data. Notably, the ScanQA dataset is the *only* dataset related to 3D in this collection. COARSE CORRESPONDENCES is applied only to the ScanQA data, while the other data maintain their original format.

**In-domain Evaluation.** We first evaluate our model on the ScanQA validation set. As shown in Table 1, our method, compared to fine-tuning on the original ScanQA without COARSE COR-RESPONDENCES, significantly enhances the model's 3D spatial understanding, even surpassing previous VLMs specifically designed for 3D tasks, which involve specialized architectural designs and are fine-tuned on much larger amounts of 3D-related data. This demonstrates that COARSE CORRESPONDENCES can also work effectively for open MLLMs.

**Out-domain Evaluation.** To further demonstrate the generalizability of our method, we conduct experiments to evaluate the zero-shot performance of our model, fine-tuned on ScanQA, on the SQA3D dataset. As shown in Table 5, on this previously unseen dataset, COARSE CORRESPONDENCES still outperforms the model fine-tuned only on the original ScanQA, proving that our method can generalize to out-of-domain datasets. Even more notably, even without using COARSE CORRESPONDENCES during inference, simply applying it during the training phase already brings

| Method | Acc |
|---|---|
| LLaVA(Fine-tuned) | 36.0 |
| LLaVA+CC(train-only) | 37.17 |
| LLaVA+CC | 39.13 |

Table 5: **Comparisons on SQA3D dataset.** COARSE CORRESPONDENCES generalizes well on out-domain dataset.

improvements. This highlights that our method is not only effective as a prompting technique for inference but also holds potential as a data augmentation method during training, which is worth further exploration in the future.

## 5 ANALYSIS

### 5.1 THE SOT BENCHMARK FOR SPATIAL ORIENTATION TEST

Considering that a crucial aspect of embodied tasks like navigation is the judgment of left-right orientation, we aimed to gain a deeper understanding of how COARSE CORRESPONDENCES influences MLLMs' comprehension of left-right spatial orientation. Specifically, we focused on two key questions: 1) Are MLLMs robust to camera motion bias? Ideally, MLLMs' understanding of left-right orientation in 3D space should be independent of whether the camera is moving from left to right or right to left, meaning MLLMs should be robust to camera motion bias. 2) Do MLLMs possess spatial perspective-taking ability, i.e., the ability to imagine how an object or scene would appear from a perspective different from the current camera viewpoint? Numerous studies in humans (Newcombe, 1989; Tversky & Hard, 2009) have shown that this ability is closely related to the development of spatial intelligence in children.

However, current benchmarks face three issues: 1) They may have been partially used in MLLM training data, 2) Current benchmarks lack annotations regarding whether the 3D space scan was conducted from left to right or right to left, making it difficult to analyze the impact of camera motion on MLLMs, and 3) Existing benchmarks evaluate a model's 3D spatial awareness from the perspective of the camera-wielding observer.

Therefore, we introduce a new diagnostic benchmark to evaluate MLLMs: Spatial Orientation Test (SOT). Once again, we show that COARSE CORRESPONDENCES improves GPT-4V,O's abilities on this new benchmark.

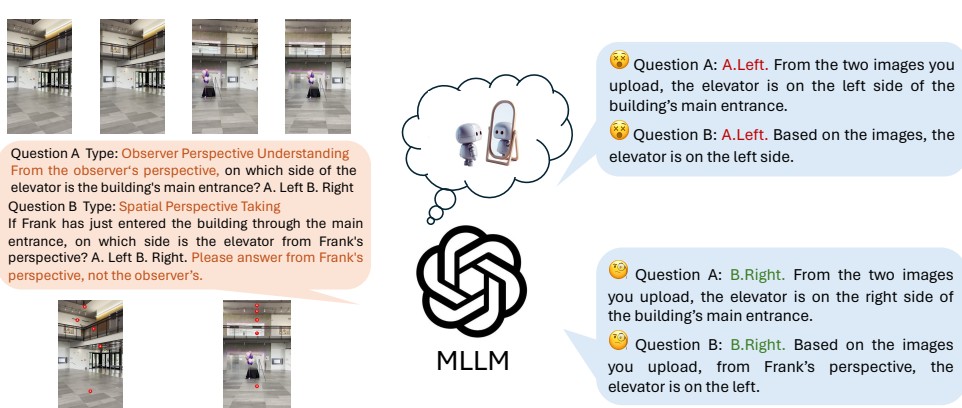

Figure 2: **Illustration of our SOT dataset.** We mention two types of questions: Observer perspective understanding and spatial-perspective taking. COARSE CORRESPONDENCES demonstrates superior effectiveness on the dataset.

**Data curation.** We manually curated ten real-world scenes, both indoor and outdoor, using different mobile devices at various viewpoints. We instructed 10 human participants to take two videos in their environment from two viewpoints. When in each viewpoint, they were asked to remain in place as they laterally pan their mobile devices to scan their 3D environment. From 20 collected scenes, we filtered to and retained 10 scenes that satisfied the following four criteria: First, we could uniquely describe one viewpoint from the perspective of the other and vice-versa. For example, in Figure 2, we define the other viewpoints as 'a person stepping out of an elevator.' Second, we ensured that no single frame captured the entire 3D space, ensuring that models can not short-cut answers using any single view. Third, all scans move the camera from left to right. Fourth, to avoid privacy concerns, we ensured that no people appeared in the videos. Each scene scan lasts between 3 to 5 seconds.

For each scene, we designed **five carefully crafted questions**, each asking the model to determine if one object is to the left or to the right of another from a specific viewpoint. The first three questions are from the observer's (camera's) perspective, while the final two describe the perspective in language, thereby, testing for a model's spatial perspective-taking ability. Human performance on these questions is 100%. We design SOT questions to have a bias towards asking about relationships between objects that appear in the first last frame of the scan, ensuring that the has to use multiple frames to answer. In total, across the 10 scenes, SOT has a modest 50 questions.

**Results.** As shown in Table 6, COARSE CORRESPONDENCES performs very well even on in-the-wild data. For instance, when using only the first and last frames, our method results in a 13.4 % improvement. As illustrated in Figure 2, our method enables GPT-4O to understand the 3D spatial structure represented by the images using minimal overlap, whereas GPT-4O alone performs only slightly better than random guessing.

More importantly, according to Table 6, we found that current MLLMs achieve significantly higher accuracy on videos filmed from left to right compared to those filmed from right to left, indicating that even models like GPT-4O have a strong camera motion bias. Our method greatly mitigates this issue. By calculating the harmonic mean of results from both left-to-right ($L->R$) and right-to-left ($R->L$) camera pans, we found that our method brought a 17.3 % improvement, indicating that COARSE CORRESPONDENCES helps MLLMs learn a more equivariant visual representation from image sequences.

Additionally, we isolated the performance on the two perspective-taking questions per scene in Figure 3. We discovered that current MLLMs still lack the ability for spatial perspective-taking. While COARSE CORRESPONDENCES improves GPT-4O's perspective-taking capability, the results are bittersweet, as they still perform worse than random guessing. This suggests that embodied spatial

| Models | Frame | Origin | Reverse | Harmonic Mean |
|--------|-------|--------|---------|---------------|
| GPT-4O | 2 | 58.2 | 50.0 | 53.8 |
| GPT-4O+CC | 2 | **71.6** | **70.6** | **71.1** |
| GPT-4O | 4 | 58.0 | 50.4 | 53.9 |
| GPT-4O+CC | 4 | **71.2** | **71.2** | **71.2** |

Table 6: **Comparisons on SOT.** COARSE CORRE-SPONDENCES shows strong capability of enhancing 3D spatial understanding of MLLMs. It can also ease the striking finding of camera motion bias of current MLLMs.

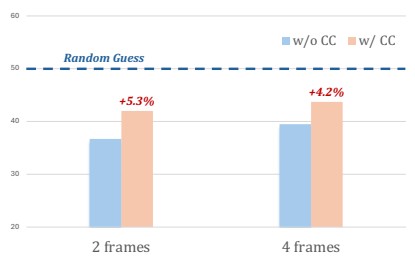

Figure 3: **Comparisons on SOT's spatial perspective-taking questions.** COARSE CORRESPONDENCES improves performance but GPT-4O still performs below random chance.

awareness has yet to emerge in MLLMs—at least for now—highlighting a potential direction for future research.

| Design Choice | B − 1 | B − 2 | METEOR | ROUGE-L | CIDER |
|---------------|-------|-------|--------|---------|-------|
| **Ablation on number of marks** | | | | | |
| 5 | **39.7** | **25.5** | **17.4** | **40.8** | **79.2** |
| 8 | 35.4 | 18.9 | 14.6 | 37.8 | 74.0 |
| **Ablation on mark size** | | | | | |
| 40px | 35.3 | 17.5 | 15.5 | 39.2 | 76.1 |
| 60px | **39.7** | **25.5** | **17.4** | **40.8** | **79.2** |
| 80px | 33.1 | 14.3 | 14.5 | 37.6 | 71.3 |
| **Ablation on mark type** | | | | | |
| markers only | 35.9 | 19.6 | 15.9 | 39.5 | 76.4 |
| + outline | **39.7** | **25.5** | **17.4** | **40.8** | **79.2** |
| + mask | 33.1 | 14.3 | 14.5 | 37.6 | 71.3 |

Table 7: **Ablations on different design choices of COARSE CORRESPONDENCES.** We studied the impact of the number, size, and type of marks on performance. All experiments were conducted on ScanQA using GPT-4V.

## 5.2 ABLATION STUDY

Here, we explore the various design decisions in our method.

**How does COARSE CORRESPONDENCES differ from other visual prompting methods?** Our proposed method calculates and highlights correspondences between images, aiming to elicit 3D and temporal understanding. Other visual prompting methods (namely Set-of-Mark (Yang et al., 2023a), 3DAxiesPrompts (Liu et al., 2023), and Chain-of-thought (Wei et al., 2023)) can also be viewed as alternative prompting methods. Given that the ground-truth answers in existing benchmarks are relatively brief, we selected a scene from ScanQA and manually designed a new question. We qualitatively compare COARSE CORRESPONDENCES against other prompting methods on this new question, as shown in Figure 4.

The orange part of Figure 4 shows our Coarse Correspondence labels are recognized by GPT-4V. The output answer provides evidence that our coarse correspondence helps GPT-4V develop a mental 3D model of the scene. Set-of-Marks provides no spatial corresponding information and therefore is unhelpful. The Axis labels in 3DAxies can be easily misrecognized by GPT-4V, leading to misleading spatial information. Though Chain-of-Thought helps identify objects, it fails to resolve the "spatial perspective-taking" issue.

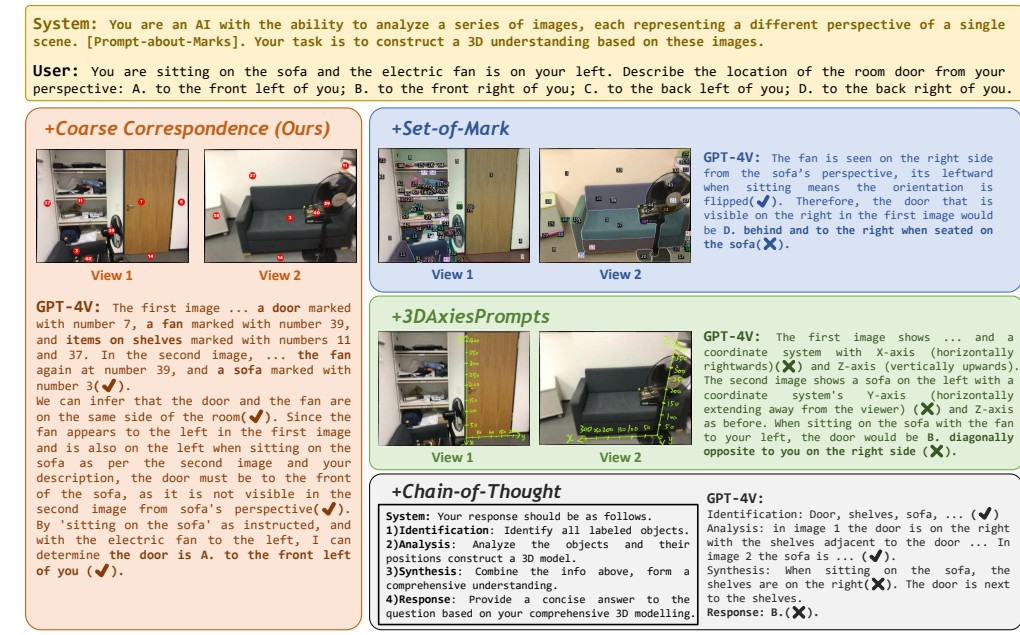

Figure 4: **Comparison of different prompting method.** Our proposed Coarse Correspondences successfully guided GPT-4V to understand 3D spatial relationships and generate the right answer. Other existing prompting method including image-based Set-of-Marks, 3DAxies and text-based Chain-of-Thought failed to answer correctly.

**Why use coarse instead of dense correspondences?** Instead of filtering and retaining only a handful of coarse correspondences, one ablation we considered is the possibility of using all dense correspondence. Unfortunately, we find that excessively overlaying too many instance marks can degrade performance (Table 7) as they occlude the visual content in the images.

**How large should the marks be?** We inject the correspondences into MLLMs by overlaying the marks into images. We empirically find an optimal mark size (where 'px' represents the mark's diameter in pixels) in Table 7. Marks that are too small tend to be ignored while those that are too large occlude visual content.

**What shape should the marks be?** We further studied the appearance of the marks. In addition to red circles with white text, we experimented with adding segmentation outlines and segmentation masks. As shown in Table 7, using segmentation outlines enhances object grounding. However, using segmentation marks occludes visual content and reduces performance.

## 6   CONCLUSION.

We propose a framework called COARSE CORRESPONDENCES prompting. By using off-the-shelf video tracking models to obtain class-agnostic, instance-level correspondences, and conveying this information to MLLMs through visual prompting, we discovered that this simple method, using only 2D images as input—without any specialized architectural design or task-specific SFT—can effectively enhance MLLMs' understanding of 3D space and long videos. This improvement extends to embodied tasks like navigation. Our method not only works on proprietary models but also generalizes to open-source models, and it performs well on both in-domain and out-of-domain datasets. Moreover, it enhances not just inference but also training. Further analysis shows that our method helps MLLMs become more robust to camera motion bias. Additionally, we identified that even GPT models struggle with perspective-taking capability, a fundamental component of human visual intelligence, which presents an important avenue for future exploration to further improve MLLMs.

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

APPENDIX

## A  BROADER IMPACT

Our method aims at improving the trustworthiness and reliability of deployment of MLLMs in real world application, including but not limited to Vision Pro, autonomous driving, and also humanoid robots. To have a virtual assistant like JARVIS in Marvel films, it's necassry to align the understanding of vision-language model with human's understanding, so that we can ensure safe application of these applications. Further, we are committed to reducing the carbon emissions produced by these models. By employing our coarse correspondence prompting method, we use a much smaller tracking module to reduce the number of input used as input to large GPT model. Besides, we also improve the speed and lower the cost of calling OpenAI API to understand a 3d scene. This enables democratize MLLMs so that more people and small companies can create their own real-world applications based on GPT-4V. We hope our work can make large AI models more effectively used for social good.

Still, we would like to point out that with the development of MLLMs, increased reliance on advanced MLLMs could also lead to a reduction in human skills, especially in interpreting and interacting with visual content. Over-dependence on these models might erode critical thinking and analytical abilities in the long term.

## B  RELATED WORK

**Multimodal language models** Multimodal LLMs(Liu et al., 2024b; Bai et al., 2023) integrate vision encoders (Radford et al., 2021) into large LLMs (Chiang et al., 2023; Touvron et al., 2023), allowing them to directly reason over visual input. Many proprietary models, such as GPT-4 (OpenAI, 2023), Gemini (Team et al., 2024), and Claude (Anthropic, 2024), as well as open-source models like the LLaVA series (Liu et al., 2024b) and BLIP series (Li et al., 2023), have made significant progress in 2D vision-language tasks like image captioning (Chen et al., 2015) and visual question answering (VQA) (Hudson & Manning, 2019; Goyal et al., 2017). Beyond these language-related tasks, many newer attempts applying MLLMs to applications such as autonomous driving (Tian et al., 2024) and robotics (Yang et al., 2023b). Many of these tasks require understanding the 3D space in which they are deployed and reason about how things are changing temporally. We improve the 3D space-time capabilities of such models.

**Visual prompting.** Effective prompting has been widely proven to improve LLMs across multiple domains. Methods, such as chain-of-thought prompting (Wei et al., 2023), force the model to reason before answering a question. For multimodal LLMs, methods such as Red-circle prompting (Shtedritski et al., 2023) and Set-of-marks (Yang et al., 2023a) can enhance the grounding abilities of CLIP (Radford et al., 2021) and GPT-4V. PIVOT (Nasiriany et al., 2024) employs visual prompting combined with iterative VQA to induce GPT-4V to generate outputs for robotics control. 3DAxies (Liu et al., 2023) enhances GPT-4V's ability to use numerical expressions to describe 3D relationships of objects in a single image by annotating a scaled 3D coordinate system on the image. Unlike these works, COARSE CORRESPONDENCES prompts MLLMs to understand the spatial relationships within a complete 3D scene from an image sequence.

**Video understanding.** Videos carry rich information about both the 3D structure as well as temporal changes in the physical world. To perform better long-horizon reasoning, work has begun incorporating video inputs into MLLMs. Recent work (Lin et al., 2023) has improved performance on video dense captioning (Krishna et al., 2017) and videoQA (Xiao et al., 2021; Grunde-McLaughlin et al., 2021). To further advance the understanding of temporal relationships in videos, EgoSchema (Mangalam et al., 2023) introduced a benchmark for long video understanding, which is more challenging than previous video-language benchmarks. Meanwhile, understanding 3D spatial relationships in videos received relatively less attention. 3D-LLM (Hong et al., 2024) converts multiview images into 3D point clouds and then feeds them into LLMs, demonstrating better results on the ScanQA (Azuma et al., 2022) benchmark for 3D understanding. OpenEQA (Majumdar et al., 2024) is also a benchmark dedicated to evaluating MLLM's understanding of 3D physical space, with outputs that are more open-vocabulary compared to ScanQA. In this paper, we propose a framework that does not require any training in modifying MLLMs; it extracts meaningful information from videos using off-the-shelf tracking models and achieves state-of-the-art performance on the benchmarks mentioned.

**Visual correspondences.** Visual correspondences have been a vital area of research in computer vision for a few decades. Applications such as Structure-from-Motion(Schonberger & Frahm, 2016) utilize correspondences to better reconstruct 3D scenes. In the past, we relied on handcrafted features like SIFT (Lowe, 2004) or SURF (Bay et al., 2006) to obtain good correspondence. Today, features extracted from deep models (Tang et al., 2023) can also provide increasingly accurate correspondences. Generally, people aim to achieve precise geometric and semantic correspondences at the pixel level. However, in this paper, we use coarse visual correspondence to prompt MLLMs, which can be easily obtained from off-the-shelf video tracking models (Yang et al., 2023c).

## C   COARSE CORRESPONDENCE IMPLEMENTATION DETAILS

As discussed in Method section, visualizing our proposed Coarse Correspondence on images will involve a centering algorithm. The inputs are selected instance segmentation masks that originally obtained from tracking model. A center of the instance mask needs to be determined in order to place the coarse correspondence marker. It is worth noting that the instance mask does not necessarily form a connected component, which makes the centering procedure worth explaining.

```python
# Find center of a mask,
# May contains multiple connected components.
def find_center(mask):
    # Go through the middle column, try to find center1
    exist_y = []
    x_center = median(left_bound, right_bound)
    for y in range(upper_bound, lower_bound):
        if (x_center, y) in mask:
            exist_y.append(y)
    if exist_y is not empty:
        y_center = median(exist_y)
        center1 = (x_center, y_center)
    else:
        center1 = None

    # Go through the middle row, try to find center2 (skip)
    if avg(center1, center2) in mask:
        return avg(center1, center2)
    elif center1 in mask:
        return center1
    elif center2 in mask:
        return center2
    else:
        center_naive = ((left_bound + right_bound)//2,
                        (upper_bound + lower_bound)//2)
        return center_naive
```

Figure 5: The pseudo code of our proposed algorithm to find the center of a given object mask. The Coarse Correspondence will be further added to the object center

As shown in the pseudo code in Figure 5, firstly we calculate the medium x-index of the masked pixels and loop through this column, trying to find the first center point. Similarly, we calculate the medium y-index of the masked pixels and loop through this row, trying to find another center point. Normally we return the average location of these two centers. If either of these centers failed to be positioned in the masked area (which may happens when the mask is not a connected components), we adopt the other one. If both of them failed to deliver, we adopt a naive center by simply averaging the four boundary.

## D   QUALITATIVE CASE STUDY

To further demonstrate the effectiveness of our proposed Coarse Correspondence under sparse image input, we defined two challenging tasks and one qualitative case study for each task.

The results of these case studies are shown in Fig. 6. Detailed illustration of the results are provided in the figure captions. The first case study is about the task of Duplicate Objects Counting, where the model needs to count the number of objects in a 3D scene. Only equipped with coarse correspondence

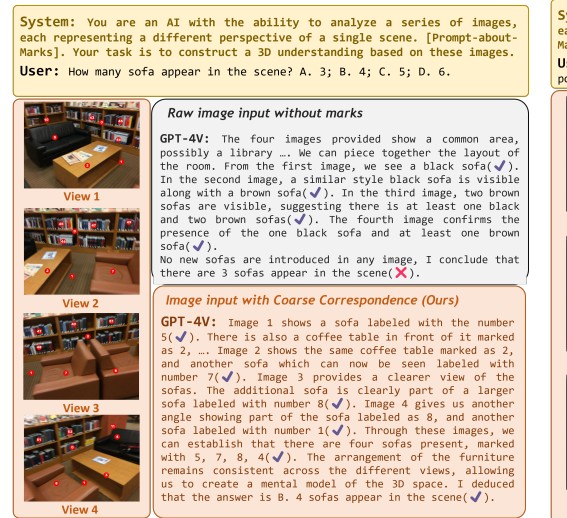

*Raw image input without marks*

**GPT-4V:** The four images provided show a common area, possibly a library …. We can piece together the layout of the room. From the first image, we see a black sofa(✓). In the second image, a similar style black sofa is visible along with a brown sofa(✓). In the third image, two brown sofas are visible, suggesting there is at least one black and two brown sofas(✓). The fourth image confirms the presence of the one black sofa and at least one brown sofa(✓).
No new sofas are introduced in any image, I conclude that there are 3 sofas appear in the scene(✗).

*Image input with Coarse Correspondence (Ours)*

**GPT-4V:** Image 1 shows a sofa labeled with the number 5(✓). There is also a coffee table in front of it marked as 2, …. Image 2 shows the same coffee table marked as 2, and another sofa which can now be seen labeled with number 7(✓). Image 3 provides a clearer view of the sofas. The additional sofa is clearly part of a larger sofa labeled with number 8(✓). Image 4 gives us another angle showing part of the sofa labeled as 8, and another sofa labeled with number 1(✓). Through these images, we can establish that there are four sofas present, marked with 5, 7, 8, 4(✓). The arrangement of the furniture remains consistent across the different views, allowing us to create a mental model of the 3D space. I deduced that the answer is B. 4 sofas appear in the scene(✓).

**System:** You are an AI with the ability to analyze a series of images, each representing a different perspective of a single scene. [Prompt-about-Marks]. Your task is to construct a 3D understanding based on these images.
**User:** You are standing facing the washbasin. Describe the relative position of the room door from your perspective.

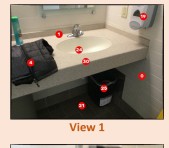
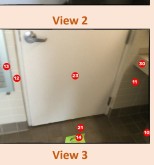

*Raw image input without marks*

**GPT-4V:** The images depict different angles of what appears to be a bathroom. In the first image, we see a washbasin ... The second image seems to be taken from an angle showing the right side of the basin ... The third image shows a door with its handle, ... (✓). From the images, it can be inferred that when facing the washbasin, the door is not in front because we see a wall with a mirror and a paper towel dispenser, not a door(✓). The first image also does not show the door to the left of the basin which suggests it's not on that side when facing the basin.(✗)... Therefore, the door must be behind the person facing the washbasin(✗).

*Image input with Coarse Correspondence (Ours)*

**GPT-4V:** From the perspectives provided, the door is most visible in Views 2 and 3, with a full front view in View 3(✓). The position of the door relative to the washbasin can be determined by the location of Object 1, which is on the washbasin's edge closest to the door, and by the visibility of the door's parts (Objects 11, 13, 12) in the different views(✓). Based on the explanations above, the door is to the left of you when you are facing the washbasin(✓).

(a) **Task: Duplicate Objects Counting.** There are 2 brown sofas and 2 black sofas. The brown sofas in View 2&4 are duplication of those in View 3. Only with the help of the Coarse Correspondence can GPT-4V understand duplicate objects between different views across a single 3D scene.

(b) **Task: Relative Location Modeling.** From View 1 & 2 we can tell that the room door is on the left-hand-side when facing the washbasin. Only with the help of the Coarse Correspondence can GPT-4V understand relative location between objects appear in different views across a single 3D scene.

Figure 6: Two complicated tasks, i.e. Duplicate Objects Counting and Relative Location Modeling are chosen to demonstrate our method. Zoom in for better view.

**System:** You are an AI with the ability to analyze a series of images, each representing a different perspective of a single scene. [Prompt-about-Marks]. Your task is to construct a 3D understanding based on these images.

**User:** You are sitting on the chair in front of the desk. Describe the relative position of these objects from your perspective: the closet, the guitar, and the bed.

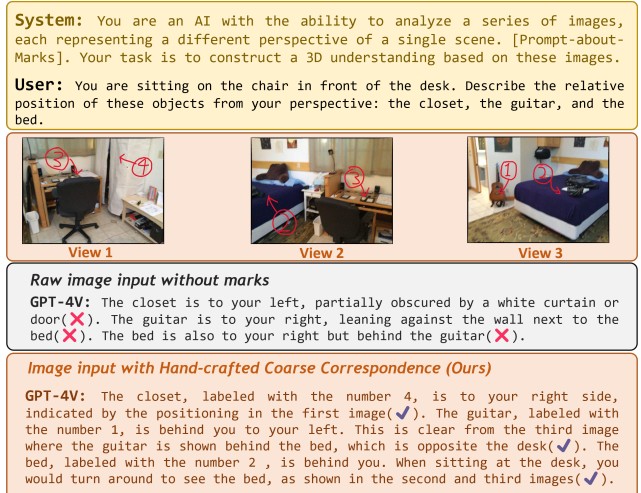

*Raw image input without marks*
**GPT-4V:** The closet is to your left, partially obscured by a white curtain or door(✗). The guitar is to your right, leaning against the wall next to the bed(✗). The bed is also to your right but behind the guitar(✗).

*Image input with Hand-crafted Coarse Correspondence (Ours)*

**GPT-4V:** The closet, labeled with the number 4, is to your right side, indicated by the positioning in the first image(✓). The guitar, labeled with the number 1, is behind you to your left. This is clear from the third image where the guitar is shown behind the bed, which is opposite the desk(✓). The bed, labeled with the number 2 , is behind you. When sitting at the desk, you would turn around to see the bed, as shown in the second and third images(✓).

Figure 7: **Hand-crafted coarse correspondence label**

can GPT-4V get a comprehensive understanding of the 3D scene, excludes the duplicate objects, and give the right answer.The second case study is about the task of Relative Location Modeling, where the model needs to understand the relative location of objects in a 3D scene. It is obvious that without the correspondence markers, GPT-4V fails to response from 3D perspective with only raw 2D images.These case studies demonstrate that our proposed Coarse Correspondence can elicit MLLMs in understanding 3D scenes from sparse image inputs.

We also prove that our Coarse Correspondence method works well with hand-crafted correspondence marks as shown in Figure 7. This further proves that our proposed method are style-agnostic as long as the marks is able to deliver the spatial correspondence knowledge.

# E More Discussions

**Limitations.** Our method relies on off-the-shelf video tracking models to obtain instance-level correspondences. Although the performance of tracking models has significantly improved with the advent of tools like SAM (Kirillov et al., 2023), achieving good results on long-form in-the-wild videos remains challenging. This is particularly evident on the 180-second EgoSchema benchmark, where Track-Anything often loses track of objects after 100 seconds, leading to inconsistent instance segmentation masks between the beginning and end of the video clip. Despite observing consistent and significant improvements on EgoSchema, we believe that accurate correspondence would further enhance the benefits of our approach.

**Relation to SlowFast** SlowFast (Feichtenhofer et al., 2019) is a framework for video recognition that includes two parallel pathways: a Slow pathway that captures motion information at a high frame rate and a Fast pathway that captures semantic information at a low frame rate. The information from both pathways is fused through lateral connections for downstream video recognition tasks. In a way, our coarse correspondence prompting can be seen as another form of SlowFast. However, unlike SlowFast, where the Slow and Fast pathways operate in parallel, our framework operates sequentially. First, it captures low-level, class-agnostic motion information at a high frame rate using a lightweight tracking model. Then, at a low frame rate, it performs recognition and reasoning requiring semantic understanding using larger MLLMs. The two stages are bridged through visual prompting. Moreover, while SlowFast learns a representation of the input video for pure vision recognition tasks such as action classification and detection, our coarse correspondence framework aims to better understand the 3D spatial structure and temporal information contained in the input video to achieve spatiotemporal perception and reasoning simultaneously.

**Eulerian vs Lagrangian** If deep learning-based methods represent camera or object motion in videos from an Eulerian viewpoint—i.e., expressing how features at fixed locations evolve over time through a multi-dimensional tensor—then our framework adds a Lagrangian viewpoint to this representation. The Lagrangian viewpoint describes the trajectories of entities moving through space and time in the video. Previously, the Lagrangian viewpoint in video descriptions has been shown to better aid human action recognition (Rajasegaran et al., 2023). Here, we demonstrate that it can more generally help MLLMs understand the 4D spatiotemporal context represented in videos.

