# OpenReview forum: "Coarse Correspondences Boost 3D Spacetime Understanding in Multimodal Language Model"
_ICLR.cc/2025/Conference — ICLR 2025 Conference Withdrawn Submission_

### Official Review · Reviewer_5KXb · 2024-11-03

**Soundness:** 3
**Presentation:** 2
**Contribution:** 3
**Rating:** 6
**Confidence:** 4

**Summary:**

This paper introduces a simple, training-free, and effective visual prompting method, COARSE CORRESPONDENCES, to improve the MLLMs' spatial and temporal understanding ability. CC uses a tracking model to find object correspondences between images, mark the same objects across the image sequence, and show huge improvements on ScanQA, OpenEQA, and EgoSchema benchmarks with state-of-the-art results. This method can also improve spatial understanding of MLLM on downstream tasks such as navigation.

**Strengths:**

1. This work points out that existing MLLMs have potential 3D spatial understanding capabilities and can be excited using simple prompt methods.
2. The author provides an effective method to enhance MLLMs' 3D and temporal understanding by simple visual prompting without complex architectural designs or downstream fine-tuning.
3. The author demonstrates the effectiveness of the method on downstream tasks like ScanQA, OpenEQA, and R2R navigation.
4. This paper presents a new benchmark, SOT, to evaluate spatial reasoning ability from alternative viewpoints.

**Weaknesses:**

1. The SOT benchmark, which evaluates spatial understanding from another viewpoint, is interesting but not very closely related to the method.
2. This method relies on the results of existing tracking models, which may introduce limitations in accuracy and robustness, particularly for long-form videos. Although existing tracking models such as SAMv2 already perform well, errors can still occur when only a few frames are sampled. The impact of these errors, as well as the solution, should be studied more.
3. Most of the author's experiments were done on closed-source models such as GPT-4V/GPT-4O. I believe that experiments on open-source models under the SAME setting would be more useful.
4. There is no section for analysis of related works, and many missing ones need to be discussed. To only mention a few:
- PointLLM: Empowering Large Language Models to Understand Point Clouds
- Point-Bind & Point-LLM: Aligning Point Cloud with Multi-modality for 3D Understanding, Generation, and Instruction Following
- An Embodied Generalist Agent in 3D World
- LLaVA-3D: A Simple yet Effective Pathway to Empowering LMMs with 3D-awareness

Some recent public ones are also encouraged to be supplemented to make a more thorough discussion.

**Questions:**

1. I'm a little confused about the experiments in the PROMPTING OPENMODELS section. Since CC is a TRAINING-FREE method, why didn't the llava experiment take the same setting as before?
2. I'm very interested in the inference time for each step of the method on the navigation task.

---

### Official Review · Reviewer_y3VQ · 2024-11-04

**Soundness:** 2
**Presentation:** 3
**Contribution:** 2
**Rating:** 5
**Confidence:** 4

**Summary:**

This work focuses on leveraging different forms of prompts to significantly enhance the understanding of 3D spatial location information by mature LLMs like GPT-4-O.

**Strengths:**

This work focuses on leveraging different forms of prompts to significantly enhance the understanding of 3D spatial location information by mature LLMs like GPT-4-O.

**Weaknesses:**

This work focuses on leveraging different forms of prompts to significantly enhance the understanding of 3D spatial location information by mature LLMs like GPT-4-O. However, I have the following questions:
1.How much improvement does this method provide for other 2D MLLMs besides those listed in the paper, such as LLAVA?
2.Besides the benchmarks mentioned in the paper, can this method be applied to more benchmarks?
3.This method seems a bit overly simplistic. Please restate its innovativeness and necessity, as well as how it differs from similar methods in the same category.

**Questions:**

This work focuses on leveraging different forms of prompts to significantly enhance the understanding of 3D spatial location information by mature LLMs like GPT-4-O. However, I have the following questions:
1.How much improvement does this method provide for other 2D MLLMs besides those listed in the paper, such as LLAVA?
2.Besides the benchmarks mentioned in the paper, can this method be applied to more benchmarks?
3.This method seems a bit overly simplistic. Please restate its innovativeness and necessity, as well as how it differs from similar methods in the same category.

---

### Official Review · Reviewer_nb3o · 2024-11-04

**Soundness:** 2
**Presentation:** 3
**Contribution:** 2
**Rating:** 3
**Confidence:** 3

**Summary:**

This paper presents a training-free visual prompting method with coarse images correspondance to enhances the 3D and temporal understanding of multimodal large language models (MLLMs). The proposed method works by identifying object correspondences across video frames or image viewpoints using a lightweight tracking model, identifying the topK most salient correspondences, and then visualizing these correspondences as visual prompts input for the MLLM for better 3D reasoning. The approach demonstrated substantial performance improvements across various benchmarks, including ScanQA, OpenEQA, EgoSchema, and R2R, outperforming state-of-the-art models in zero-shot settings. The author also curate a diagnostic dataset called the Spatial Orientation Test (SOT) to assess the models' ability to perform spatial perspective-taking from viewpoints other than the camera's. The results demonstrate that Coarse Correspondences significantly improves MLLMs' performance on these tasks, establishing new state-of-the-art results.

**Strengths:**

1. **Simplicity and Effectiveness**:
The method is training-free, leveraging existing tracking models to create visual prompts, making it straightforward and efficient. By overlaying unique markers on frequently occurring objects, the method supplies explicit spatial cues that aid the model's reasoning capabilities. It can also be applied broadly to various tasks without the need for task-specific finetuning or additional training of the MLLMs.
2. **Significant Performance Gains**:
The method boosts performance on multiple 3D and temporal understanding tasks across proprietary and open-source MLLMs.
3. **Reduced Computational Load**:
Coarse visual correspondence requires fewer resources to process compared to dense correspondence, which can be computationally intensive and may overwhelm the model with excessive data. By focusing on key correspondences, the proposed method maintains a balance between performance improvement and computational efficiency.
4. **Introduction of the SOT Dataset**:
The curated Spatial Orientation Test (SOT) dataset provides a valuable resource for evaluating spatial perspective-taking, a challenging aspect of spatial reasoning for MLLMs. The dataset points out current limitations and areas for future improvement.

**Weaknesses:**

1. **Dependence on Tracking Model Quality:**
The effectiveness of Coarse Correspondences highly depends on the accuracy of the lightweight tracking model used to establish object correspondences. Errors or biases in the tracking model could negatively impact the MLLM's understanding, leading to incorrect or misleading reasoning.

2. **Scope Limitation**:
Object-level correspondence may limit the model's overall 3D understanding when tasks require a deeper subject-level understanding and reasoning, such as interactions involving complex subjects (e.g., opening a drawer or detailed human interactions).
Additionally, the SOT dataset comprises only 10 scenes with a total of 50 questions, which may not be sufficient to fully assess spatial perspective-taking abilities.

3. **Visual Occlusion Issues**:
The addition of coarse correspondences and visual markers may not fully resolve visual occlusion challenges, where key visual information is partially or entirely blocked. This could limit model comprehension, especially when dealing with dense or complex scenes. Empirically find an optimal mark size may not work in some cases and rely on manual effort.

4. **Assumption of Prominent Object Importance:**
The method focuses on the most frequent object instances, which may overlook less frequent but contextually significant objects.
This could result in a biased understanding of the scene, neglecting critical elements necessary for accurate reasoning.

5. **Lack of In-depth Analysis on Failure Cases**:
The paper does not extensively cover the limitations or scenarios where coarse correspondences technique may not work as expected.

**Questions:**

- **Suggestion:**
The paper presents a simple yet effective approach to enhancing 3D spatial and temporal understanding in multimodal large language models through the use of Coarse Correspondences. However, notable weaknesses include the method's reliance on the quality of the tracking models, potential visual clutter introduced by overlaying markers, and the limited scope of the method and the dataset. These concerns prevent me from giving a positive evaluation at this moment. Resolving the aforementioned issues would strengthen the submission.

- **Additional questions:**
How scalable is the method when applied to very long video sequences or datasets with extensive temporal changes? Does the sampling approach used in coarse correspondences retain sufficient contextual information for models to perform well on such datasets, or are there limitations that need addressing when handling longer temporal contexts?

---

### Note · Authors · 2024-11-15

I have read and agree with the venue's withdrawal policy on behalf of myself and my co-authors.